# IS CLIP FOOLED BY OPTICAL ILLUSIONS?

**Jerry Ngo**
MIT CSAIL*
ngop@mit.edu

**Swami Sankaranarayanan**
MIT CSAIL
swamiviv@mit.edu

**Phillip Isola**
MIT CSAIL
phillipi@mit.edu

## ABSTRACT

Recent large machine learning models such as CLIP have shown impressive generalization performance for various perception tasks. In this work, we explore to what extent they model the human cognitive process. We focus our attention on how these models perceive optical illusions. We present a simple way to assess the effect by presenting illusions in the form of image and text prompts while observing the changes in models' output under different illusory strengths. Our results show that CLIP can indeed be fooled by different types of illusions relating to lightness and geometry.

## 1 INTRODUCTION

Foundation models (Bommasani et al., 2021) are a class of large over-parameterized models that accomplish various tasks relating to human perception. In this work, we analyze one such large vision-language model, CLIP (Radford et al., 2021), which has been shown to possess properties similar to human cognition such as multimodal neurons that encode concepts across vision and language, understanding emotion composition (Goh et al., 2021), exhibiting Stroop effect (Radford et al., 2021) and sound symbolism (Near, 2022). We consider CLIP as a visual system and probe it with a variety of optical illusions which have been known to pose a challenge to human cognition.

There are several previous works in machine learning exploring optical illusions from the perspective of low-level vision (Kim et al., 2019), (Sun & Dekel, 2021). However, all previous works only use the visual modality to measure the effect. In this work, we explore a new way to probe the ability of machine learning models to capture illusions by using multimodal stimuli: image and text.

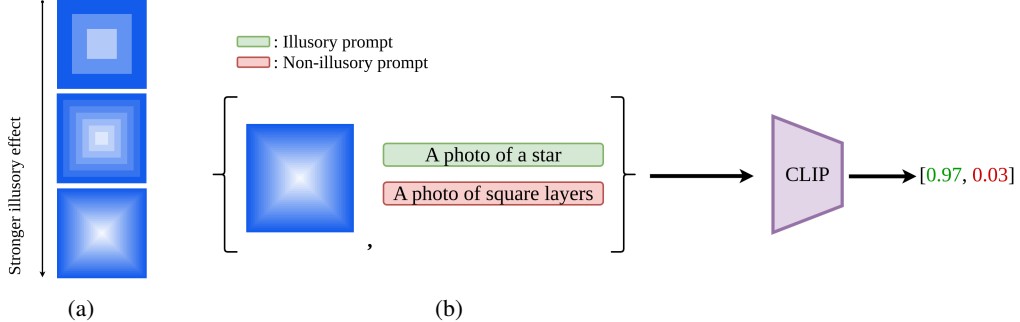

Figure 1: As an example, we use the Vasarely illusion, in which a perceived glowing star appears as the number of superimposed squares increases. (a) Varying the illusory strength from weak to strong. (b) For a pair of illusory and non-illusory prompts, to assess how CLIP perceives an illusion, we classify illusory images to one of those two prompts by comparing the output CLIP scores.

## 2 ILLUSION CLASSIFICATION USING CLIP

Given an image $I$ and a prompt $T$, we represent the CLIP similarity as $\text{CLIP}(I, T)$. For each illusion, we have $N$ text prompt pairs of $(p, q)$ where $p$ describes the illusory signal (illusory prompt) and

---

*This work was done when visiting CSAIL.

$q$ ignores the illusion (non-illusory prompt). Given an illusory stimulus $X$ of an illusion, the CLIP score for illusory and non-illusory image-text pair can be given as:

$$S_{ill}^i, S_{non-ill}^i = \frac{1}{N} \sum_i^N \text{softmax}(\text{CLIP}(X, [p_i, q_i]))$$  (1)

For an image of an optical illusion, if CLIP scores the illusory prompt higher, i.e., $S_{ill}^i > S_{non-ill}^i$ then it implies that the input image is classified as an illusion by CLIP. Our scoring approach is illustrated in Figure 1.

## 3 EXPERIMENT AND RESULTS

To assess CLIP's perception of optical illusions, we prepared a set of 11 optical illusions that include variations from weak to strong illusory effects. Our goal is to determine whether changes in CLIP's score correlate with the strength of the illusion. To achieve this, we plot CLIP's score $\{S_{ill}, S_{non-ill}\}$ as a function of illusory strength. As we increase the illusory strength in the input image, if CLIP perceives these illusions as humans do, we expect an upward trend in illusory prompts' classification score and vice versa for non-illusory prompts. To ensure that the CLIP output is normalized for prompt variability, we averaged the CLIP response across multiple prompts for each illusory strength value. These multiple prompts are paraphrases of the original prompt (specified by the authors) that were generated by ChatGPT (OpenAI, 2023). In addition, we introduced a calibration method based on Zhao et al. (2021) to de-bias CLIP, which is described in the Appendix.

Figure 2 illustrates how CLIP's response changes with varying levels of illusory strength of three illusions that successfully deceive CLIP. In general, we observe an upward trend for the illusory prompts and a downward trend for the non-illusory prompts. For instance, in the Vasarely illusion, the first few images do not exhibit an illusory effect on the human eye, and CLIP accurately tracks this. Similarly, for the other two illusions, we observe a progression from weaker to stronger variations (which all show some illusory effect) that CLIP successfully anticipates. However, for the remaining eight illusions, CLIP either predicts all images as illusory prompts without any change in prediction score or predicts all images as non-illusory prompts. In both cases, these illusions don't fool CLIP, and details about these results are provided in the Appendix.

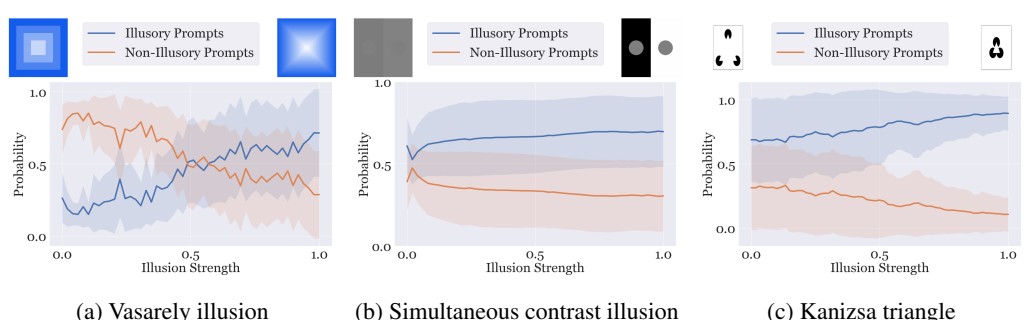

(a) Vasarely illusion     (b) Simultaneous contrast illusion     (c) Kanizsa triangle

Figure 2: For each illusion, the orange line indicates the mean classification score between visual stimuli and illusory prompts across different pairs of prompts, while the blue line indicates the mean classification score of non-illusory prompts. The filled area between two lines denotes the value within 1 standard deviation from the mean.

## 4 CONCLUSION

In our study, we proposed a novel approach to evaluate the perception of a large foundation model, CLIP, on optical illusions using multimodal data. Our findings show that out of the 11 illusions tested, CLIP can be reliably fooled by 3 of them. This is an intriguing result that merits future research into the robustness of large models like CLIP and the underlying mechanisms that make machine learning models susceptible to such inputs.

URM STATEMENT

The authors acknowledge that at least one key author of this work meets the URM criteria of ICLR 2023 Tiny Papers Track.

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

# A APPENDIX

## A.1 OPTICAL ILLUSIONS STIMULI

Optical illusions are a phenomenon where the perceived visual world is different from reality. Studying illusions can lead to interesting observations as it can reveal how our brain processes visual information (Adelson, 1999). The existence of optical illusions has long intrigued scientists across diverse fields (Buckle et al., 2013). Below of the list of 11 included optical illusions in this paper and their generation procedure.

Table 1: List of Illusion Stimuli

| Illusion Name | Number of Images |
|---|---|
| Vasarely illusion | 50 |
| Ponzo illusion | 23 |
| Simultaneous contrast illusion | 50 |
| Café wall illusion | 8 |
| Scintillating grid illusion | 15 |
| Kanizsa triangle | 61 |
| Delboeuf illusion | 16 |
| Ebbinghaus illusion | 5 |
| Muller-Lyer illusion | 31 |
| Zollner illusion | 76 |
| Rod and frame illusion | 20 |

### A.1.1 VASARELY ILLUSION

This illusion comprises superimposed squares with ordered levels of lightness. As the number of squares increases, an X-like shape appears along the diagonal of the image. This X does not exist.

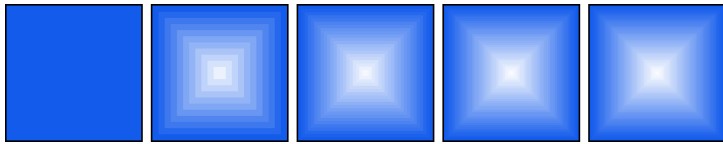

Figure 3: Vasarely illusion

We generated illusion photos with different amounts of squares. For lightness, we used the HSL color model to modify the color of each imposed square. We varied the lightness (L) values from 50 to 100. If an image contains $n$ square, the lightness values used in that image are the set $\{\lfloor 50 + \frac{50a}{n} \rfloor \,|\, 0 \le a < n, a \in N\}$. Accordingly, we had the amounts of imposed squares going from 1 to 50. All images are in the $1:1$ ratio with a size of 336 to 400 pixels. All the images are resized to 336 pixels by a pre-processing function before being passed into CLIP. 5 images sampled from the generated stimuli are depicted in Figure 3.

### A.1.2 PONZO ILLUSION

This illusion depicts a rail/ladder from the lower end's point of view. On the rail, there are two horizontally parallel lines of the same length. However, the line in the lower end seems to be shorter.

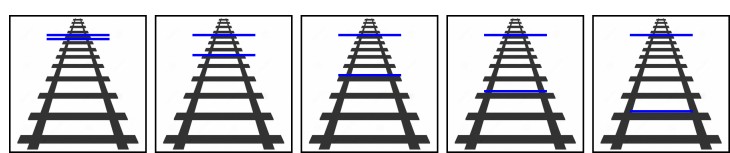

Figure 4: Ponzo illusion

We generated the Ponzo illusion by using a $336 \times 336$ image of a rail. Both horizontal lines have the same length of 226 pixels. The upper line is fixed to 45 pixels from the top. The lower line's location varies from 275 pixels to 30 pixels from the top, and every location jump is 10 pixels apart. In total, we generated 23 images for this illusion.

### A.1.3 SIMULTANEOUS CONTRAST EFFECT

This is a phenomenon where the surrounding luminance affects how the inner luminance is perceived. In this work, we focus on a simple version of this illusion. In this variation, the illusion is a square image divided into two areas with different luminance levels. Each area has a circle at the center, and both circles have the same luminance level. The circles in the darker area appear to be lighter and vice versa.

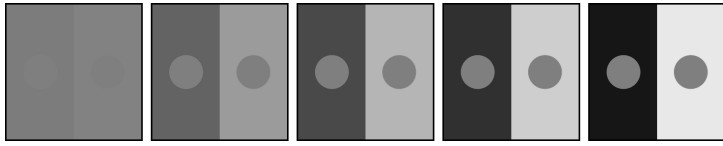

Figure 5: Simultaneous contrast illusion

As the difference between the surroundings and the inner circle increase, the illusion effect also increases. Thus, we could test how CLIP interprets the effect by varying the brightness of both canvases. The canvases are created using the Pyllusion package from Makowski et al. (2021). Using the HSL color system, we fixed the two circles with the color code of $[160, 0, 50]$. The left area uses the same color, with the $L$ values varying from $51$ to $100$. Whereas the right area has $L$ values varying from $49$ to $0$. Combining two areas together, we got a total of 50 photos of different pairs of brightness.

### A.1.4 CAFÉ WALL ILLUSION

The illusion consists of a checkboard-like pattern. Even though the board is made up of parallel lines of black and white tiles, those lines appear to be sloped.

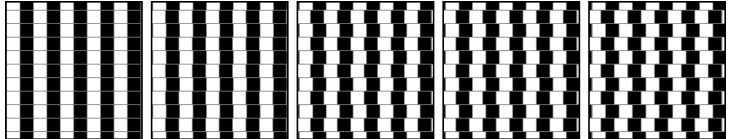

Figure 6: Café wall illusion

We generated all the stimuli with the size of $341 \times 374$. Each image comprises 10 equal-size black-and-white parallel lines stacked upon each other with 2-pixel mortar lines in between. In these lines, we used the RGB values of $[255, 255, 255]$ for white squares, $[0, 0, 0]$ for black squares, and the median of these two color codes for mortar lines. In the first image of the illusion, all parallel lines are aligned. Using this image, we generated the second image in which every odd line (first, third...) is shifted 1 pixel to the right and all other lines are shifted 1 pixel to the left. Using the same principle of shifting the previous photo's parallel lines, we generated 8 photos for the Café wall illusion.

### A.1.5 SCINTILLATING GRID ILLUSION

The illusion depicts a grid of black squares separated by orthogonal gray bars. At the intersection of gray lines, there are discs of white color. However, these white discs rapidly changed to white at random locations.

We generated stimuli for this illusion using sine functions. The amplitude of these functions started with a value of 3.0 and gradually declined to 0 with a 0.2 decrease after each photo. These made up a collection of 15 $336 \times 336$ images. The black background color has the $[0, 0, 0]$ RGB code, while the discs and mortar lines use the standard white and gray color from the Matplotlib.

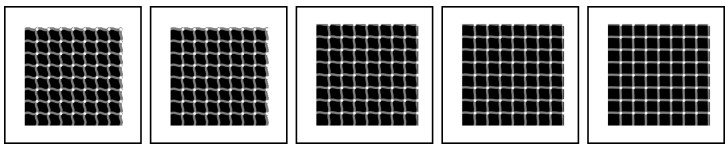

Figure 7: Scintillating grid illusion

### A.1.6 KANIZSA TRIANGLE

Kanizsa triangle is an image of three sliced circles arranged in the formation of a triangle. Even though there is no explicit contour between these circles, we perceive a white triangle by illusory contour.

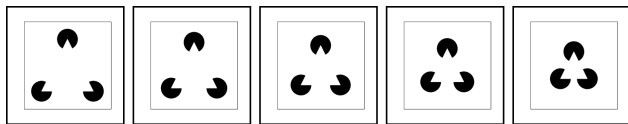

Figure 8: Kanizsa triangle

We use the python library kanizsa (Zi, 2021) to generate this illusion. The settings we used to generate these images are $radius = 40, width = 336, height = 336, center = [168, 168]$. We varied the distance between the three black sliced circles by changing the $distance$ parameter from 20 to $-40$ to get a total of 61 images of $432 \times 288$.

### A.1.7 DELBOEUF ILLUSION

The illusion consists of two circles of the same sides positioned side by side. One of the circles is inside a bigger black ring, making this circle seem smaller.

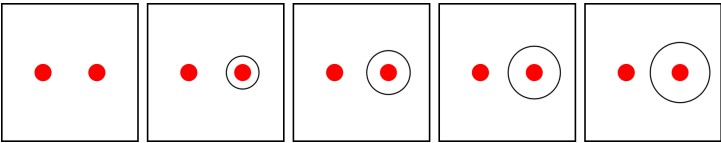

Figure 9: Delboeuf illusion

We used the Pyllusion library to generate this illusion. We set $distance = 0.8$ and varied the $illusion\_strength$ parameter from 0 to 15 to obtain 16 $400 \times 400$ images.

### A.1.8 EBBINGHAUS ILLUSION

The setup of this illusion is similar to the Delboeuf illusion, in which we both have two circles next to each other. There are circles surrounding both main circles, and the one with bigger black circles seems smaller.

We used the Pyllusion library to generate this illusion. We varied the $illusion\_strength$ parameter from 0 to 4 to obtain 5 $400 \times 400$ images.

### A.1.9 MULLER-LYER ILLUSION

The Muller-Lyer has multiple variations, but we focus on the one with two line segments. There are two line segments of the same length placed near each other. From both ends of each segment, there are arrowhead-like shapes. These arrowheads could point inward or outward from one segment, but the line segmented with outward-pointing heads seems shorter.

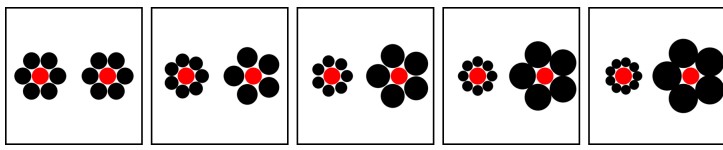

Figure 10: Ebbinghaus illusion

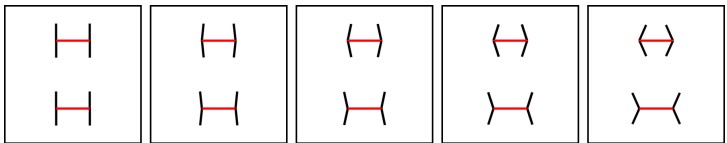

Figure 11: Muller-Lyer illusion

We used the Pyllusion library to generate this illusion. We varied the $illusion\_strength$ parameter from 0 to 30 to obtain 31 $400 \times 400$ images.

### A.1.10 ZOLLNER ILLUSION

In the Zollner illusion, we have a set of parallel lines. These lines are repeatedly crossed by shorter diagonal lines, making them no longer parallel.

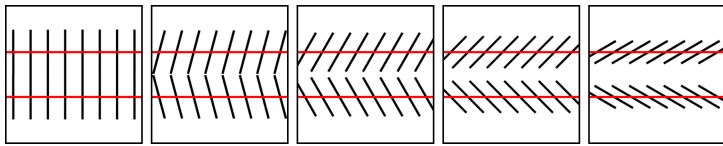

Figure 12: Zollner illusion Collection

We used the Pyllusion library to generate this illusion. We varied the $illusion\_strength$ parameter from 0 to 75 to obtain 76 $400 \times 400$ images.

### A.1.11 ROD AND FRAME ILLUSION

The rod and frame illusion consists of a vertical line inside a square frame. However, as the frame starts to tilt, the perception of the line being vertical is affected.

We used the Pyllusion library to generate this illusion. We varied the $illusion\_strength$ parameter from 0 to 19 to obtain 20 $400 \times 400$ images.

### A.2 OTHER OPTICAL ILLUSIONS' RESULTS

The remaining results are depicted in Figure 14.

### A.3 CLIP CALIBRATION

Given an image and a text prompt, CLIP provides a score indicating the similarity between the semantic content of the image and text prompt. Using this raw score to classify an image into one of many prompts can be an effective tool when the concepts represented in the prompts have a clear semantic meaning, such as objects or animals. However, in our case, the prompts describe more abstract concepts, and utilizing raw scores can result in a biased prediction. For example, it could be unclear if the CLIP score is high for an illusion image and prompt pair due to CLIP being fooled by the illusion or if there is a random bias unaccounted for in the text prompt.

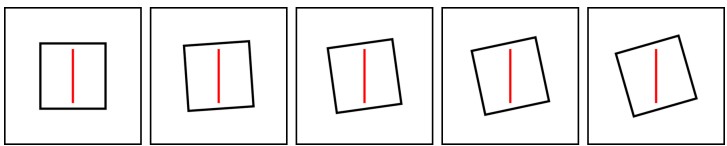

Figure 13: Rod and frame illusion collection

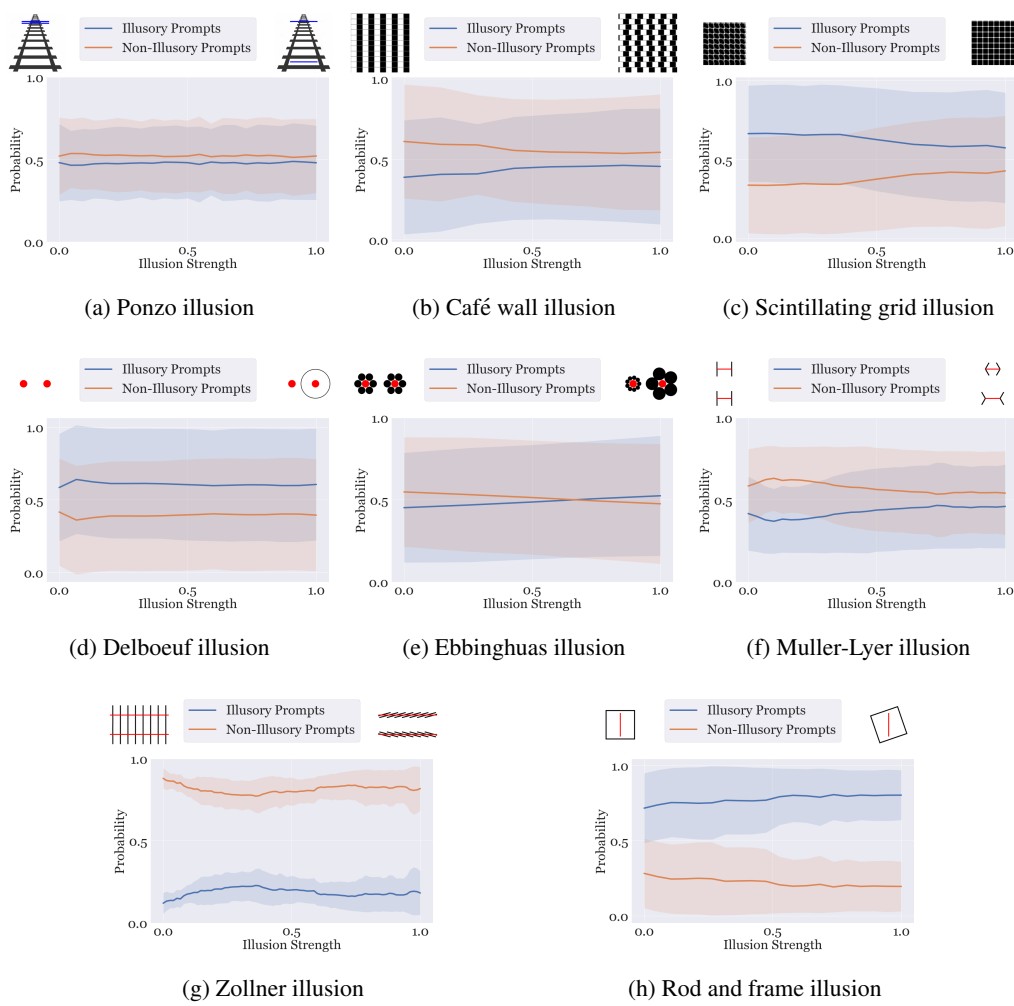

Figure 14: Illusion and non-illusory prompts classification scores' trend for other illusions

To remove this bias, we follow the method described in Zhao et al. (2021) which discusses how large language model (LLM) results could provide unstable results depending on the prompt format, in-context sample choice, and permutation due to factors like training data distribution. They propose to obtain a content-free prediction to calibrate LLM's output, which helps reduce variance and improve performance. We apply a similar approach to calibrate the CLIP scores.

Let $I$ be an image and $T$ be a text prompt that describes the image. We define $\tilde{s} = \text{softmax}(\text{CLIP}(I, T))$ as the uncalibrated similarity score. Now let $\{I_c^{(i)}\}_{i=1}^N$ be a set of $N$ random natural images that will be used for calibration. We define $\bar{s} = \frac{1}{N} \sum_{i=1}^{N} \text{softmax}(\text{CLIP}(I_c^{(i)}, T))$ as the average CLIP score for each prompt. Finally, we get a calibrated score $s = \text{softmax}(\tilde{s}/\bar{s})$. For our experiments, $\{I_c^{(i)}\}_{i=1}^N$ is a set of 100 images randomly drawn from the ImageNet dataset's training set. Unless specified, all the comparison scores in this paper are calibrated.

### A.4 CHATGPT PROMPT GENERATION

Using text to describe the illusion and then calculating the distance between visual and textural illusory stimuli has the advantage of quickly setting up new tasks for different illusions. However, we could describe an illusion in many different ways. For example, in the Delboeuf illusion, you can describe the images as "a small ball and a big ball" or "a big red dot and a small red dot." This poses the problem of what is the best pair of prompts to use for one illusion. To solve this issue, we collected many pairs of prompts and took the mean of their classification score.

As the perceived effect and the actual physical properties of an illusion have been described multiple times in vision science, we could find a way to paraphrase and modify these descriptions to create different pairs of prompts of the same semantic. We made use of ChatGPT to do such a task.

#### A.4.1 GENERATION PROCEDURE

For each illusion, we provide a fixed instruction with examples of four illusory prompts and four non-illusory prompts as follows:

> Below are annotations for the same image. There are two sets of annotations. The first set of annotation describes the image, and the second one doesn't, which contrast the first annotation, so we can use them for the classification task:
> First set:
> - first illusory prompt
> - second illusory prompt
> - third illusory prompt
> - fourth illusory prompt
> Second set:
> - first non-illusory prompt
> - second non-illusory prompt
> - third non-illusory prompt
> - fourth non-illusory prompt
> Generate two sets of the image with ten annotations for each set. Try to keep the length of annotations within each set as similar as possible.

For a few illusions, we get the list of 10 illusory prompts and 10 non-illusory prompts in the first trial. However, for the majority of the illusion, there must be some additional instructions, and regeneration needs to be done after the above instruction. For example, sometimes, ChatGPT would alter the color description of the shape in the illusion or generate less than 10 prompts. In that case, we would provide instructions such as "don't change the ball color" or "generate 10 more prompts" and then select generated prompts to get a set of 10 illusory and 10 non-illusory prompts for each illusion. One exception here is the Zollner illusion where ChatGPT wasn't able to stick to the illusion description and we had to only use 5 generated prompts pair for this illusion.

#### A.4.2 LIST OF PROMPTS

**Vasarely illusion**

Sample prompts:

<table>
<tr><td>Illusory Prompts</td><td>Non-Illusory Prompts</td></tr>
</table>

Illusory Prompts

- A photo of a star on blue background
- A set of stacked boxes with a diagonal bright X shape
- An image with a star shape in the center
- A photo of stacked blue squares with a star in the middle

Non-Illusory Prompts

- A photo of stacked squares
- A set of layered boxes ranging from deep blue to light blue
- Squares of different sizes and brightness
- A set of layered boxes ranging from deep blue to light blue

Generated prompts:

Illusory Prompts

- An image of a bright star on a blue background with stacked boxes in the foreground
- A photo of a blue background with a prominent star and stacked boxes in the corner
- A picture of stacked blue boxes with a centered star on a blue background
- A photo of a star with stacked blue boxes in the background against a blue background
- A blue background with a star in the middle and stacked boxes in the foreground
- An image with a blue background, a bright white star in the center, and stacked boxes around it
- A photograph of a star on a blue background with boxes stacked around it
- A picture of a bright star in the center with stacked blue boxes surrounding it
- A photo of a blue background with a prominent star and stacked boxes in the background
- An image of a blue background with a star in the center and stacked boxes on either side

Non-Illusory Prompts

- A photo of stacked squares with varying sizes and colors
- A picture of boxes in shades of blue, stacked one on top of another
- An image of square boxes in various shades of blue, arranged in a stack
- A photo of blue boxes stacked on top of each other, creating a gradient effect
- A picture of a stack of boxes in varying shades of blue
- An image of blue boxes arranged in a stack with different shades of blue
- A photo of stacked boxes in shades of blue, creating a color gradient
- A picture of boxes stacked upon each other with shades of blue ranging from deep to light
- An image of a stack of blue boxes, with a gradient effect created by the different shades
- A photo of boxes in shades of blue, arranged in a stack with varying sizes and brightness

**Ponzo illusion**

Sample prompts:

| Illusory Prompts | Non-Illusory Prompts |
| --- | --- |
| • A photo of a rail with two blue lines on it, one is shorter than the other | • A photo of a rail with two blue lines on it, both with the same length |
| • A photo of a big blue line on top of a rail and another small blue line at the bottom | • A photo of a blue line on top of a rail and a blue line at the bottom and two lines have the same size |
| • A photo of a train rail where two horizontal blue dashes of different sizes lay on | • A photo of a train rail where two horizontal blue dashes of the same size lay on |
| • A photo of a train rail where two parallel blue dashes of different sizes lay on the rail | • A photo of a train rail where two parallel blue dashes of the same size lay on the rail |

Generated prompts:

| Illusory Prompts | Non-Illusory Prompts |
| --- | --- |
| • A photo of a rail with two blue lines on it, where one line is shorter than the other | • A photo of a rail with two blue lines on it, where both lines have the same length |
| • An image depicting a large blue line positioned on top of a rail, accompanied by a smaller blue line at the bottom | • An image depicting a blue line on top of a rail and another blue line at the bottom, both lines of equal size |
| • A photo capturing a train rail with two horizontal blue dashes of different sizes resting on it | • A photo capturing a train rail with two horizontal blue dashes of the same size resting on it |
| • A photograph of a metal rail with two blue stripes of unequal length painted on it | • A photograph of a metal rail with two blue stripes of equal length painted on it |
| • A picture of a train track with two blue lines running across it, one line being shorter than the other | • A picture of a train track with two blue lines running across it, both of the same length |
| • A photo of a railway line featuring two horizontal blue dashes of different lengths | • A photo of a railway line featuring two horizontal blue dashes of equal length |
| • A picture of a railway line with two blue lines running parallel to each other, but one is shorter than the other | • A picture of a railway line with two blue lines running parallel to each other, both of the same length |
| • A photo of a train rail with two distinct blue stripes of different lengths | • A photo of a train rail with two distinct blue stripes of the same length |
| • A photo displaying a rail with two blue lines, where one line is longer than the other | • An image of a railway track with two blue dashes painted on it, both of equal length |
| • An image featuring a blue line on top of a rail and a smaller blue line at the bottom, both with different lengths | • A picture of a metal rail with two blue stripes painted on it, with both lines of the same length |

**Simultaneous contrast illusion**

Sample prompts:

| Illusory Prompts | Non-Illusory Prompts |
|---|---|
| • A photo is divided into two rectangles with two circles: the circle in the dark rectangle is lighter and the circle in the light rectangle is darker | • A photo is divided into two rectangles with two circles: the circle in the dark rectangle and the circle in the light rectangle has the same luminance |
| • A photo of a light gray circle to the left of the dark gray circle | • A photo of a gray circle to the left of a gray circle, both with the same luminance |
| • A photo of two circles: one is darker, one is lighter. The light gray canvas contains the darker circle and the dark gray canvas contains the lighter circle | • A photo of two circles of the same shade of gray. The light gray canvas contains a gray circle and the dark gray canvas contains a gray circle |
| • A photo of two circles with two different shades of gray | • A photo of two circles with the same shades of gray |

Generated prompts:

| Illusory Prompts | Non-Illusory Prompts |
|---|---|
| • A photo of two circles on a gray background, one circle is lighter than the other | • A photo of two circles on a gray background, both circles have the same luminance level |
| • A photo of two circles on a gray background, one circle is darker than the other | • A photo of two circles on a gray background, both circles have the same shade of gray |
| • A photo of two gray circles with different luminance levels | • A photo of two gray circles with identical luminance levels |
| • A photo of two circles, one in a lighter gray shade and one in a darker gray shade, on a gray canvas | • A photo of two circles of the same size and same gray shade, positioned on a gray canvas |
| • A photo of two circles, one in a light gray shade and one in a dark gray shade, positioned on a split background with the light gray on the left | • A photo of two circles of the same shade of gray, positioned on a split background with the same gray shade on both halves |
| • A photo of a circle in a light gray shade positioned in the left half of the image, and a circle in a darker gray shade positioned on the right half of the image | • A photo of a circle in a gray shade positioned in the left half of the image, and a circle of the same gray shade positioned on the right half of the image |
| • A photo of two circles positioned on a gray background, with the same gray shade but different luminance levels | • A photo of two circles positioned on a gray background, with the same gray shade and the same luminance levels |
| • A photo of two gray circles of the same size but different shades of gray | • A photo of two gray circles of the same size and same shade of gray |
| • An image depicting two circles: one lighter circle on the dark gray canvas and one darker circle on the light gray canvas | • A photo of two circles of the same size and same shade of gray, positioned on a gray canvas, with one circle in the left half of the image and the other circle in the right half of the image |

- A photo of two circles, one in a lighter gray shade and one in a darker gray shade, on a gray canvas, with the lighter circle in the left half of the image

- A photo of two circles of the same shade of gray, positioned on a split background with the same gray shade on both halves, with one circle in the left half of the image and the other circle in the right half of the image

**Café wall illusion**

Sample prompts:

| Illusory Prompts | Non-Illusory Prompts |
| --- | --- |
| • A photo of a black and white board, black and white tiles made of tilting lines | • A photo of a black and white board, black and white tiles made of parallel lines |
| • A photo of a checkerboard with oblique lines | • A photo of a checkerboard with parallel lines |
| • A photo of diagonal stacked lines of interleaving black and white lines | • A photo of aligned, stacked lines of interleaving black and white lines |
| • A photo of sloped lines making up a black and white chessboard | • A photo of parallel lines making up a black-and-white chessboard |

Generated prompts:

| Illusory Prompts | Non-Illusory Prompts |
| --- | --- |
| • A photo of a black and white checkerboard with diagonal lines | • A photo of a black and white board, black and white tiles made of parallel lines |
| • A photo of a chessboard with diagonally-oriented black and white squares | • A photo of a checkerboard with parallel lines |
| • A photo of an alternating pattern of black and white diagonal lines | • A photo of aligned, stacked lines of interleaving black and white lines |
| • A photo of a board with tilted black and white squares arranged in a checkerboard pattern | • A photo of parallel lines making up a black-and-white chessboard |
| • A photo of a diagonal pattern made of black and white lines on a board | • A photo of a board with a pattern of straight black and white squares arranged in a checkerboard |
| • A photo of black and white squares arranged in a tilted checkerboard pattern | • A photo of black and white squares with parallel lines in a checkerboard pattern |
| • A photo of a board with a pattern of oblique black and white lines | • A photo of an array of parallel black and white lines making up a checkerboard pattern |
| • A photo of a tilted chessboard with black and white squares | • A photo of a chessboard with straight black and white squares arranged in parallel lines |
| • A photo of an array of slanted black and white tiles forming a checkerboard | • A photo of a board with straight black and white squares arranged in a grid-like pattern |
| • A photo of a tilted pattern of black and white squares forming a chessboard | • A photo of a pattern of black and white parallel lines forming a chessboard |

**Scintillating grid illusion**

Sample prompts:

Illusory Prompts

- A photo of black blobs grid and at the intersections of these blobs, there are black and white circles
- A photo of a black grid with vertical and horizontal gray lines, at the crossing of these lines are black and white circles
- A photo of black and white circles at points of intersection
- A photo of black and white circles at crossing points

Non-Illusory Prompts

- A photo of black blobs grid and at the intersections of these blobs, there are white circles
- A photo of a black grid with vertical and horizontal gray lines, at the crossing of these lines are white circles
- A photo of white circles at points of intersection
- A photo of white circles at crossing points

Generated prompts:

Illusory Prompts

- An image showing a black grid with circular patches of black and white at the intersections of the gridlines
- A picture of black and white circles arranged in a grid pattern
- A photo of a checkerboard pattern made up of circles, some black and some white
- A visual of a grid of black blobs with circles of alternating black and white at their intersections
- A picture of a pattern of circular patches in which the black and white circles appear to merge in places
- An image of a grid of black and white circles with black borders at their intersections
- A photograph of a grid of circular patches, in which the circles alternate between black and white and overlap slightly
- A picture of a black grid with small circular patches of black and white at the intersections of the gridlines
- An image showing a grid of circular patches, with black and white circles at the intersections that appear to blend together
- A photo of a pattern of circular patches, some black and some white, arranged in a grid-like formation

Non-Illusory Prompts

- An image showing a black grid with circular patches of white at the intersections of the gridlines
- A picture of white circles arranged in a grid pattern
- An image showcasing a black grid with vertical and horizontal gray lines, featuring white circles at the crossing points
- A visual of a grid of black blobs with circles of white at their intersections
- A picture of a pattern of circular patches in which the white circles appear to merge in places
- An image of a grid of white circles with black borders at their intersections
- A photo illustrating the existence of white circles at the points of intersection within a black blobs grid
- A picture of a black grid with small circular patches of white at the intersections of the gridlines
- An image showing a grid of circular patches, with white circles at the intersections that appear to blend together
- A photo capturing white circles arranged at the intersections of a grid composed of black blobs

**Kanizsa triangle**

Sample prompts:

Illusory Prompts

- A photo of a white triangle in the middle and three black circles at each vertex of the triangle
- A photo of three circles being obscured by a white triangle
- A photo of three sliced black circles making up a triangle in the center
- A photo of a white triangle accompanied by three solid circular shapes

Non-Illusory Prompts

- A photo of three circles equally spaced from each other
- A photo of three circles, one on top, two at the bottom
- A photo of three sliced black circles on a white background
- A photo of three black circles

Generated prompts:

Illusory Prompts

- A photo of a white triangle with black circles at each vertex on a neutral background
- An image featuring a triangle composed of three black circles with a white background
- A photo of a white triangle surrounded by three solid black circular shapes
- An image depicting a triangle formed by three black circles intersecting on a white background
- A photo of a white triangle with three black circles at its corners against a plain backdrop
- A composition featuring a white triangle and three black circles positioned at the vertices
- An image of a white triangle with black circular elements at each vertex against a solid backdrop
- A photo showcasing a triangle made up of three black circles partially obscured by a white overlay
- A photo of a white triangle accompanied by three solid circular shapes
- A photo of three black circles forming a triangle with a white triangle in the center

Non-Illusory Prompts

- A photo of three equidistant circles on a neutral background
- An image featuring a single circle positioned above two circles in a row
- A composition of three sliced black circles against a plain backdrop
- A photo of three circles evenly spaced from each other on a neutral background
- An image depicting a single circle positioned above two circles below on a white backdrop
- A composition featuring three sliced black circles against a neutral background
- An image of three circles equally spaced from each other against a plain background
- A photo of three circles, one on top, two at the bottom
- A photo of three sliced black circles on a white background
- A photo of three black circles on a white background

**Delboeuf illusion**

Sample prompts:

| Illusory Prompts | Non-Illusory Prompts |
|---|---|
| • A photo of a small red circle and a big red circle | • A photo of two red circles, both of the same size |
| • A photo of a big red circle in a small black circle and a small red circle in a big black circle | • A photo of two equal red circles, positioned side by side |
| • A photo of a bigger red circle on the left and a smaller red circle on the right | • A photo of a red circle in a small black circle and a red circle in a big black circle; both red circle share the same size |
| • A photo of two red circles of different sizes, positioned side by side | • A photo of a red circle on the left and a red circle on the right, both have the same size |

Generated prompts:

| Illusory Prompts | Non-Illusory Prompts |
|---|---|
| • A photograph of two red circles, one larger than the other, positioned side by side | • A photograph of two equal-sized red circles placed side by side |
| • A picture of a small red circle and a big red circle, next to each other | • An image of two identical circular shapes next to each other |
| • An image showing two red circles of different sizes, placed side by side | • A photo of two red circles of equal size positioned side by side on a white background |
| • A photo of a large red circle and a small red circle, placed adjacent to each other | • A picture of two equal-sized circular shapes, positioned adjacent to each other |
| • A picture of two circular shapes, one bigger and one smaller, positioned next to each other | • An image of two circular shapes of the same size, placed next to each other |
| • A photograph of two red circles with different sizes, side by side on a white background | • A photograph of two identical red circles positioned side by side |
| • An image of two circular shapes, one larger than the other, positioned next to each other | • A picture of two red circles of equal size, placed side by side |
| • A picture of two red circles of different sizes, one bigger and one smaller, placed next to each other | • A photo of two circular shapes of the same size placed adjacent to each other |
| • A photograph of two adjacent red circles, one larger and one smaller in size | • An image of two equal-sized red circles placed next to each other |
| • An image of two red circles of different sizes, placed side by side | • A photograph of two circular shapes of equal size placed side by side |

**Ebbinghaus illusion**

Sample prompts:

|  |  |
|---|---|
| **Illusory Prompts** | **Non-Illusory Prompts** |

**Illusory Prompts**

- A photo of a small red circle and a big red circle

- A photo of a big red circle in a small black circle and a small red circle in a big black circle

- A photo of a bigger red circle on the left and a smaller red circle on the right

- A photo of two red circles of different sizes, positioned side by side

**Non-Illusory Prompts**

- A photo of two red circles, both of the same size

- A photo of two equal red circles, positioned side by side

- A photo of a red circle in a small black circle and a red circle in a big black circle; both red circle share the same size

- A photo of a red circle on the left and a red circle on the right, both have the same size

Generated prompts:

**Illusory Prompts**

- A photograph of two red circles, one larger than the other, positioned side by side

- A picture of a small red circle and a big red circle, next to each other

- An image showing two red circles of different sizes, placed side by side

- A photo of a large red circle and a small red circle, placed adjacent to each other

- A picture of two circular shapes, one bigger and one smaller, positioned next to each other

- A photograph of two red circles with different sizes, side by side on a white background

- An image of two circular shapes, one larger than the other, positioned next to each other

- A picture of two red circles of different sizes, one bigger and one smaller, placed next to each other

- A photograph of two adjacent red circles, one larger and one smaller in size

- An image of two red circles of different sizes, placed side by side

**Non-Illusory Prompts**

- A photograph of two equal-sized red circles placed side by side

- An image of two identical circular shapes next to each other

- A photo of two red circles of equal size positioned side by side on a white background

- A picture of two equal-sized circular shapes, positioned adjacent to each other

- An image of two circular shapes of the same size, placed next to each other

- A photograph of two identical red circles positioned side by side

- A picture of two red circles of equal size, placed side by side

- A photo of two circular shapes of the same size placed adjacent to each other

- An image of two equal-sized red circles placed next to each other

- A photograph of two circular shapes of equal size placed side by side

**Muller-Lyer illusion**

Sample prompts:

<table>
<tr><td align="center">Illusory Prompts</td><td align="center">Non-Illusory Prompts</td></tr>
<tr><td>

- A photo of a long red line and a short red line, positioned horizontally

- A photo of two red lines, one red line is longer

- A photo of two red lines, the one on top is shorter than the one on the bottom

- A photo of a red line placed on top of another red line, both red line have different length

</td><td>

- A photo of two equal red lines, positioned horizontally

- A photo of two red lines, both of the same length

- A photo of two red lines of the same length, the one on top and the one on the bottom

- A photo of a red line placed on top of another red line; both red line have the same size

</td></tr>
</table>

Generated prompts:

<table>
<tr><td align="center">Illusory Prompts</td><td align="center">Non-Illusory Prompts</td></tr>
<tr><td>

- A photograph of two parallel red lines where one is longer than the other
- A photograph of two red lines that are parallel to each other with different lengths
- A picture of two red lines that are parallel to each other, with the top line being shorter than the bottom line
- An image of two red lines placed parallel to each other where one line is shorter than the other
- A photo of two red lines running parallel to each other, but with different lengths
- A photo of two red lines, one longer than the other, and both running parallel to each other
- An image of two red lines, one on top of the other, with different lengths
- An image of a long and short red line, placed in parallel next to each other
- An image of two lines in red, one long and one short, placed parallel to each other
- A photo of two red lines, positioned in parallel, with one line longer than the other

</td><td>

- A photograph of two parallel red lines with the same length

- A photograph of two red lines that are identical and parallel to each other

- A picture of two parallel red lines, one on top and one on the bottom, that have the same length

- An image of two red lines placed parallel to each other where both lines have the same length

- A photo of two red lines of the same length, running parallel to each other

- An image of two equal-length red lines running in parallel

- A photo of two equal-length red lines, positioned in parallel next to each other

- An image of two parallel red lines that have the same length

- A photo of two lines in red, with equal length, appearing parallel to each other

- An image of two identical red lines, placed in parallel to each other

</td></tr>
</table>

**Zollner illusion**

Sample prompts:

| Illusory Prompts | Non-Illusory Prompts |
|---|---|
| • A photo of two tilting red lines | • A photo of two parallel red lines |
| • A photo of two horizontally tilting red lines | • A photo of two horizontally parallel red lines |
| • A photo of two horizontally tilting red lines with multiple black lines crossed over them vertically | • A photo of two horizontally parallel red lines with multiple black lines crossed over them vertically |
| • A photo of two horizontally slanting red lines | • A photo of two horizontally aligned red lines |

Generated prompts:

| Illusory Prompts | Non-Illusory Prompts |
|---|---|
| • A photo of two red lines slanted at an angle on a white background | • A photo of two parallel red lines with the same thickness on a white background |
| • A photo of two red lines forming a V-shape, tilted on a white background | • A photo of two straight red lines, parallel to each other on a white background |
| • A photo of two red lines tilted at an angle with black lines intersecting them on a white background | • A photo of two red lines parallel to each other, with thin black lines crossing them, on a white background |
| • A photo of two slanted red lines with the same thickness on a white background | • A photo of two red lines with the same thickness and same length, parallel on a white background |
| • A photo of two red lines with a thin black line intersecting them, tilted on a white background | • A photo of two red lines with equal thickness, parallel to each other on a white background |

**Rod and frame illusion**

Sample prompts:

| Illusory Prompts | Non-Illusory Prompts |
| --- | --- |
| • A photo of a tilted red rod | • A photo of a vertical red rod |
| • A photo of a tilted red rod inside a black square frame | • A photo of a vertical red rod inside a black square frame |
| • A photo of a slanted red rod | • A photo of an upright red rod |
| • A photo of a slanted red rod inside a black square frame | • A photo of a red upright rod inside a black square frame |

Generated prompts:

| Illusory Prompts | Non-Illusory Prompts |
| --- | --- |
| • A photo of a tilted red bar | • A photo of a vertical red bar |
| • A photo of a tilted red bar inside a black rectangular frame | • A photo of a vertical red bar inside a black rectangular frame |
| • A photo of a slanted red bar | • A photo of an upright red bar |
| • A photo of a slanted red bar inside a black rectangular border | • A photo of a red vertical bar inside a black rectangular border |
| • A photo of a red stick leaning diagonally | • A photo of a red stick standing vertically |
| • A photo of a red rod positioned diagonally | • A photo of a red stick standing upright |
| • A photo of a red bar placed at an angle | • A photo of a red rod placed vertically |
| • A photo of a red pole tilted at an angle | • A photo of a red pole positioned vertically |
| • A photo of a red column slanted at an angle | • A photo of a red column standing vertically |
| • A photo of a red pillar leaning diagonally | • A photo of a red pillar positioned vertically |

