# OpenReview forum: "Is CLIP Fooled by Optical Illusions?"
_ICLR.cc/2023/TinyPapers — Submitted to Tiny Papers @ ICLR 2023_

### Official Review · Reviewer_uYRp · 2023-03-25

**Confidence:** 4

**Summary Of Contributions:**

This paper evaluates CLIP's ability to perceive optical illusions by presenting them in the form of image and text prompts and observing changes in the model's output under different levels of illusion strength. The results demonstrate that certain types of illusions can fool CLIP.

**Rating:**

High Potential (HP): a submission which meets the reviewing criteria and has potential to make an impact on the field

**Strengths And Weaknesses:**

Strengths:
- This paper studies an interesting problem: whether CLIP can be fooled by optical illusions. It reveals some limitations of CLIP, which can be quite useful in some cases especially considering CLIP is widely adopted in many scenarios.
- The experiments are extensive (regarding CLIP) and the authors have provided detailed descriptions in the appendix.
- The writing is overall clear and easy to follow.

Weaknesses:
- Overall I quite like this paper, but I would suggest the authors slightly weaken the claim from evaluating the "foundation models" to "CLIP" alone. As the whole paper is about CLIP and some other multimodal models may have different ways of accepting input or different architectures which may lead to different behaviors. E.g., in the conclusion section: "we proposed a novel approach to evaluate the perception of large foundation models".
- How many images are used for each illusion? I didn't find it in the main text or appendix. Am I missing something? If there is only a small number of images are used for evaluation, the results may be unreliable.

**Suggested Changes:**

- To attract more interest, the authors may consider discussing in which real-world scenarios the limitation of CLIP can have an impact, such as cases where the visual inputs have similar lightness and geometry patterns as the optical illusions.
- Can be future work: it would be beneficial to analyze why CLIP fails to recognize the specific illusions.

---

### Official Review · Reviewer_NA6k · 2023-03-26

**Confidence:** 3

**Summary Of Contributions:**

This paper explores whether the CLIP (Contrastive Language–Image Pre-training ) model is "fooled" by certain optical illusions in a similar manner to humans. It measures this by presenting a set of 11 illusions as image inputs, with sets of illusory and non-illusory captions, and measuring which captions CLIP predicts better fit the images. Several of the illusions are found to successfully "fool" CLIP.

**Rating:**

Clear, Correct, and Reproducible (CCR): a submission which meets the reviewing criteria

**Strengths And Weaknesses:**

Strengths
* The work probes what I think is a very interesting question - do CLIP models exhibit certain "flaws" of human vision due to being trained with human caption pairs?
* The paper is nicely written and the figures are clear. It's a satisfying read for 2 pages. The appendices are also useful and I appreciate the inclusion even of the negative results and the presentation of them in Fig 14.
* The method, including the calibration approach, seems good apart from a few weaker points mentioned below.

Weaknesses
* Creation of additional captions. For some of the generated captions, there does not seem much difference between the illusory and non-illusory, e.g. for the Rod and frame illusion, "A photo of a red column standing vertically" was generated as a non-illusion prompt, but seems to perfectly describes the illusion prompt as well. Should some human filtering be used to help "clean" the generated captions?
* CLIP is trained on 400 millions text-image pairs. For famous illusions, it seems there is a good chance of many examples of them leaking into the CLIP training set. I think this is alluded to in the paper although it could be made more explicit in the main text: is this why it's important to use captions which wouldn't have been used previously to describe the images in the CLIP dataset?
* The purpose of the study could be made a bit more explicit. Is it to investigate the potential flaws that training on human-given image-caption pairs might induce in a model trained on those pairs? Or is it to understand whether similar illusory experiences can arise in a general vision system like CLIP, despite it never having seen any similar illusory images before? I think it's good to be explicit here, because I think the paper is doing the former, while for the latter, it seems like it would require that CLIP not be trained on any similar illusory images?

**Suggested Changes:**

In addition to adding a few details to address the weaknesses above, here are some minor typos / comments -
* "Foundation models (Bommasani et al., 2021) are a class of large over-parameterized models that accomplish various tasks relating to human perception. " -- I think this is true of vision foundation models, but not sure it describes other foundation models such as LLMs well?
* In Sec 2., it could be made clearer what an illusion vs non-illusion prompt are - just by adding the example referring back to Fig 1. I guess illusion prompt = a photo of a star, while non-illusion prompt  = photo of square layers, but this is not totally obvious on first read.
* In Fig 1, it might be good to make it more obvious to the reader what the illusion is about: seeing an "X", and making it very clear that this perceived "X" does not exist in pixel space -- I'm not sure that the Vasarely illusion is that widely known.
* in Fig 2, it would be nice to add the actual prompts
* CLIP should be referenced the first time it is mentioned in the introduction (cite Radford et al) so that unfamiliar readers can follow better
* "Zhao et al. (2021)" citation formatting is funny , should be "(Zhao et al., 2021)"

---

### Author Response · Authors · 2023-06-01
**Changes in Revised Paper**


To all reviewers, thank you so much for all your constructive feedback. We have made the following changes to our papers:

-   Replaced the term *“illusion/non-illusion prompt/strength”* with *“illusory/non-illusory prompt/strength”* across the paper

-   Added a brief description of the Vasarely illusion to Figure 1 for readers who are not familiar with the illusion

-   Added labels for illusionary/non-illusionary prompts to Figure 1 so Section 2 would be clearer

-   Weakened the claim and conclusion from evaluating “foundation models” to “CLIP”

-   Add a table to list the number of images for each illusion in the appendix

-   Fixed multiple format errors and typos.

- Filtered and regenerated a few prompts within some illusions. However the number of updated prompts are minimal which results the same trends within each illusion.

Once again, thank you for your invaluable contributions to our paper. Your suggestions have greatly contributed to improving the quality and clarity of our work.

---

### Meta-Review · Area_Chair_qYQQ · 2023-04-07

**Recommendation:** Invite to present
**Confidence:** 3

**Metareview:**

The paper explores an interesting problem by evaluating CLIP's ability to perceive optical illusions. The experiments are extensive, and the writing is overall clear and easy to follow. There are weaknesses in the paper's claim, the number of images used for each illusion is not mentioned, and the purpose of the study could be made more explicit.

**Summary:**

The paper evaluates CLIP's ability to perceive optical illusions by presenting them in the form of image and text prompts and observes changes in the model's output under different levels of illusion strength.

**Comments And Feedback To The Authors:**

- The authors may consider discussing in which real-world scenarios the limitation of CLIP can have an impact, such as cases where the visual inputs have similar lightness and geometry patterns as the optical illusions.
- The paper should add more details to address some of the weaknesses pointed out by the reviewers, such as creating additional captions and clarifying the purpose of the study.
The authors should also address some minor typos and formatting issues.


**Reason For Not Giving A Higher Recommendation:**

The submission meets the reviewing criteria, but some minor issues need to be addressed, such as the authors should slightly weaken their claim from evaluating "foundation models" to "CLIP" alone, and the number of images used for each illusion is not mentioned.

**Reason For Not Giving A Lower Recommendation:**

The paper reveals some significant limitations of CLIP, which can be quite useful.

---

### Decision · Program_Chairs · 2023-04-09

Invite to present